# Increased Biomass and Polyhydroxybutyrate Production by *Synechocystis* sp. PCC 6803 Overexpressing *RuBisCO* Genes

**DOI:** 10.3390/ijms24076415

**Published:** 2023-03-29

**Authors:** Vetaka Tharasirivat, Saowarath Jantaro

**Affiliations:** Laboratory of Cyanobacterial Biotechnology, Department of Biochemistry, Faculty of Science, Chulalongkorn University, Bangkok 10330, Thailand

**Keywords:** RuBisCO, *Synechocystis* sp. PCC6803, PHB, glycogen, intracellular lipids

## Abstract

The overexpression of the *RuBisCO* (*rbc*) gene has recently become an achievable strategy for increasing cyanobacterial biomass and overcoming the biocompound production restriction. We successfully constructed two *rbc*-overexpressing *Synechocystis* sp. PCC 6803 strains (OX), including a strain overexpressing a large subunit of RuBisCO (OX*rbcL*) and another strain overexpressing all large, chaperone, and small subunits of RuBisCO (OX*rbcLXS*), resulting in higher and faster growth than wild type under sodium bicarbonate supplementation. This increased biomass of OX strains significantly contributed to the higher polyhydroxybutyrate (PHB) production induced by nutrient-deprived conditions, in particular nitrogen (N) and phosphorus (P). As a result of higher PHB contents in OX strains occurring at days 7 and 9 of nutrient deprivation, this enhancement was apparently made possible by cells preferentially maintaining their internal lipids while accumulating less glycogen. The OX*rbcLXS* strain, with the highest level of PHB at about 39 %w/dry cell weight (DCW) during 7 days of BG_11_-NP treatment, contained a lower glycogen level (31.9 %w/DCW) than wild type control (40 %w/DCW). In contrast, the wild type control strain exposed to N- and NP-stresses tended to retain lipid levels and store more glycogen than PHB. In this model, we, for the first time, implemented a *RuBisCO*-overexpressing cyanobacterial factory for overproducing PHB, destined for biofuel and biomaterial biotechnology.

## 1. Introduction

A part of global biomass and atmospheric oxygen are reliably produced by cyanobacteria, which are photosynthetic prokaryotes capable of fixing atmospheric CO_2_ and converting it to cellular carbon supply [1,2,3]. In terms of sustainable and renewable applications, cyanobacteria have established themselves as one of the most promising bioresources for alternate biofuels and bioproducts, even though their yield and efficiency have a tight correlation with their biomass [4,5]. To overcome the lower carbon constraint to the productivity of biofuel and bioproducts, many approaches for increasing algal biomass have been developed. Promising approaches to achieve this objective include nutritional adjustment or the genetic and metabolic engineering of cyanobacteria and algae. With respect to biofuel applications, the high content of stored carbohydrate means that microalgae are great candidates for bioethanol production [6,7,8,9,10]. Use of a nitrate-deprived medium for culturing the cyanobacterium *Synechococcus* sp. PCC 7002 resulted in a certain reduction in cell growth, but increased total carbohydrate content, herein glycogen, resulting in a higher C:N ratio of hydrolyzed biomass that could be further utilized by yeast and produced more ethanol [9]. The genetic engineering approach has also been applied to generate a higher biomass solution. Native overexpression of *rbc* or the *RuBisCO* gene operon encoding the ribulose-1,5-bisphosphate carboxylase/oxygenase (RuBisCO) enzyme in the Calvin–Benson–Bassham (CBB) cycle showed enhanced growth and photosynthesis in *Synechocystis* sp. PCC 6803 [11]. Each co-overexpression of the four genes encoding four enzymes in the CBB cycle of *Synechocystis* sp. PCC 6803, including RuBisCO, fructose-1,6/sedoheptulose-1,7-bisphosphatase (FBP/SBPase), transketolase (TK), and aldolase (FBA), with *pdc* and *adh* genes encoding pyruvate decarboxylase and alcohol dehydrogenase, respectively, produced higher ethanol and total biomass [5,11]. Moreover, *Synechocystis* with co-overexpression of *RuBisCO* and *glpD*, encoding glycerol-3-phosphate dehydrogenase, produced significantly higher contents of intracellular lipids and secreted free fatty acids (FFAs) [12]. Heterologous expression of the *Synechococcus elongatus* PCC 6301 *rbcLS* gene, encoding large and small subunits of RuBisCO, in *Synechococcus elongatus* PCC 7942, enabled cells to increase isobutyraldehyde production [13]. In addition, *S. elongatus* PCC 7942 RuBisCO expression in *Synechococcus* sp. PCC 7002 was able to induce free fatty acid (FFA) production [14]. A higher level of 2,3-butanediol bioproduct was also obtained by engineered *S. elongatus* PCC 7942 with improved glucose utilization and increased CO_2_ fixation in the CBB cycle [15].

The metabolic networks of the CBB cycle and the biosynthetic pathways of glycogen, polyhydroxybutyrate (PHB), fatty acids, and membrane lipids are shown in Figure 1. Bicarbonate (HCO_3_^−^) transporters normally transport bicarbonate into the cyanobacterial cell to develop into a carbon-concentrating mechanism in the subcellular compartment, herein called the carboxysome [16]. Three bicarbonate (HCO_3_^−^) uptake systems in cyanobacteria consist of BCT1, SbtA, and BicA [17,18]. For diffused CO_2_, there are two thylakoid-bound CO_2_ uptake systems, including NDH-I_3_ and NDH-I_4_, which convert CO_2_ into HCO_3_^−^ [16]. The carbonic anhydrase (CA) enzyme, encoded by *CcaA*, in the carboxysome dehydrates HCO_3_^−^ to CO_2_, which is fixed by RuBisCO in the RuBP carboxylation reaction to generate 3-phosphoglycerate (3PGA). Triose phosphates derived from 3PGA are used by the CBB cycle to regenerate RuBP or may be utilized further in central carbon catabolism. The glyceraldehyde-3-phosphate (GAP) intermediate can either yield pyruvate or enable regeneration of the RuBP precursor in the CBB cycle. The Embden–Meyerhof–Parnas (EMP) pathway starts from GAP, converting subsequently to fructose-6-phosphate (F6P) and glucose-6-phosphate (G6P), which connect with glycogen metabolism [19]. The formation of ADP-glucose from G6P is catalyzed by ADP-glucose pyrophosphorylase (glgC), and glycogen synthesis occurs by elongating the glucose chain, which is catalyzed by glycogen synthase (glgA) [20,21,22]. Glycogen is degraded by glycogen phosphorylase (glgP) or glycerol-3-phosphate dehydrogenase (glgX) [19]. During nitrogen starvation, glycogen, a major carbon storage molecule, is catabolized and converted into polyhydroxybutyrate (PHB) [23,24]. Acetyl-CoA is a main precursor for generating energy via the TCA cycle and producing PHB, fatty acids, and membrane lipids. The PHB synthetic pathway derives from an acetyl-CoA precursor catalyzed by β-ketothiolase (phaA), acetoacetyl-CoA reductase (phaB), and heterodimeric PHB synthase (phaE and phaC), respectively [25,26]. The TCA cycle is the main pathway for acetyl-CoA to produce cellular energy, whereas fatty acid and lipid syntheses are also crucial for membrane integrity. For fatty acid synthesis, acetyl-CoA is converted to malonyl-CoA by acetyl-CoA carboxylase (ACC) and flows through the FAS II system to produce an intermediate fatty acyl-ACP, which is generally used as a precursor for membrane lipid synthesis catalyzed by several acyltransferase enzymes (PlsX and PlsC) [27,28]. Moreover, membrane lipids may be hydrolyzed by the lipase A enzyme (LipA) to generate free fatty acids (FFAs), which are either recycled to enhance fatty acyl-ACP by acyl-ACP synthetase (AAS) or secreted out into the medium [12].

In this study, to supply more carbon from the CBB cycle, we aimed to overexpress the native *RuBisCO* operon genes (or *rbc*), including large (L), chaperone (X), and small (S) subunits, in *Synechocystis* sp. PCC 6803. Two engineered strains consisted of (1) *Synechocystis* sp. PCC 6803, overexpressing the large subunit of RuBisCO or OX*rbcL*, and (2) *Synechocystis* sp. PCC 6803, overexpressing all subunits of RuBisCO or OX*rbcLXS*. The cell growth of OX strains was significantly increased when bicarbonate was added into BG_11_ medium. In particular, this enhanced carbon supply could induce cells to produce more PHB when faced with stressed conditions. Both OX strains produced higher PHB content by about 31–39 %w/DCW with lower glycogen accumulation under nutrient-deprived conditions, herein BG_11_-N and BG_11_-NP, when compared to the wild type control.

## 2. Results

### 2.1. Overexpression of Native rbc Genes in Synechocystis sp. PCC 6803

First, three engineered *Synechocystis* sp. PCC 6803 strains, including wild type control (WTc), OX*rbcL*, and OX*rbcLXS* (Table 1), were constructed by double homologous recombination. The wild type control (WTc) strain was created by substituting the *psbA2* gene with a *Cm^R^* cassette in the *Synechocystis* wild type (WT) genome (Figure 2A). For single and triple recombinant plasmids, a native *rbcL* (or *slr0009*) gene fragment with a size of 1.4 kb, and a triple gene fragment containing *rbcL-rbcX* (or *slr0011*)-*rbcS* (or *slr0012*), with a size of about 2.4 kb, were ligated between flanking regions of the *psbA2* gene of the expression vector pEERM and the upstream region of *Cm^R^* cassette, respectively (Table 1, Figure 2B,C). By PCR, using specific primer pairs (Appendix A), all overexpressing (OX) strains were confirmed for their complete segregation and gene location (Figure 2B,C). PCR products with rbcL_F and CM_R primers confirmed the correct sizes of 2.3 and 3.3 kb, respectively, in OX*rbcL* and OX*rbcLXS* strains (Figure 2B.1,C.1), compared with no band in WT and WTc. The UppsbA2_F and DSpsbA2_R primers confirmed the expected sizes of about 3.6 and 4.6 kb in OX*rbcL* and OX*rbcLXS*, respectively (Figure 2B.2,C.2), while there were 2.4 and 2.2 kb in WT and WTc, respectively. Moreover, RT-PCR data confirmed gene overexpression in all OX strains (Figure 2D). The oxygen evolution rate of both OX*rbcL* and OX*rbcLXS* was also increased when compared to WT and WTc (Figure 2E).

### 2.2. Growth, Intracellular Pigment Contents and PHB Accumulation under Bicarbonate Supplementation

As a carbon source, NaHCO_3_ (sodium bicarbonate) supplementation is substantially used for cyanobacterial cell growth instead of CO_2_ gas bubbling [30], resulting in a pool of HCO_3_^−^, which is subsequently supplied with more CO_2_ in the carboxysome [17]. We found a higher growth rate and shorter doubling time of both OX*rbcL* and OX*rbcLXS* under sodium bicarbonate supplementation when compared to WTc, whereas there were no significant changes under the normal BG_11_ condition (Figure 3A,B). Likewise, higher levels of intracellular pigments, including chlorophyll a and carotenoids, were noted in OX strains than in WTc under the sodium-bicarbonate-supplemented condition (Figure 3C–F).

For bioplastic PHB production, higher accumulation occurred at about 7.30 and 6.40 %w/DCW in OX*rbcL* and OX*rbcLXS*, respectively, than in WTc with 4.65 %w/DCW of PHB content at day 11 under normal growth conditions (Figure 4A). It is worth noting that the sodium-bicarbonate-supplemented BG_11_ medium (BG_11_ + NaHCO*_3_*) could accelerate higher PHB production earlier in all strains at days 3 and 7 of cultivation (Figure 4B). On the other hand, the lipid content of cells grown under the BG_11_ + NaHCO_3_ condition was slightly higher than for those cells grown in normal BG_11_ medium (Figure 4C,D). However, glycogen accumulation was lowered in all strains under the BG_11_ + NaHCO_3_ condition (Figure 4E,F). In contrast, the increased glycogen production was noted at day 15 of cultivation under the normal BG_11_ condition, representing the stationary phase of cells grown in nutrient-depleted medium (Figure 4E).

### 2.3. Growth and PHB Accumulation of Cells Treated under Nutrient-Deprived Conditions

In this study, all *Synechocystis* strains were grown for 7 days in BG_11_ + NaHCO*_3_* medium before being transferred to nutrient-deprived medium (Figure 5). During stressed treatment, the nutrient-deprived conditions, including BG_11_-N and BG_11_-NP, caused a certain decrease in growth rate and chlorophyll a content, except for the phosphorus-deprived condition (BG_11_-P). In contrast, striking increases in PHB content were noted in OX strains treated under BG_11_-N and BG_11_-NP conditions, compared to WTc (Figure 6). The highest level of PHB was found in the OX*rbcLXS* strain, with 38.9 and 38.1 %w/DCW at day 7 of treatment under BG_11_-NP and BG_11_-N, respectively (Figure 6B,D). In addition, the PHB production of OX*rbcLXS* at day 9 of BG_11_-NP treatment was continuously induced up to 39.7 %w/DCW, followed by OX*rbcL*, which contained 34.6 %w/DCW of PHB (Figure 6D). Nile-red-stained cells showed higher amounts of PHB granules in OX strains than WTc at day 7 of BG_11_-NP treatment (Figure 6E). On the other hand, total lipid contents were lowered in all strains under the BG_11_-P condition, whereas insignificant amounts appeared under the BG_11_, BG_11_-N and BG_11_-NP conditions (Figure 7A). The unsaturated lipid contents showed slight increases when compared to those under normal BG_11_ condition (Figure 7B). For glycogen accumulation, a high content of glycogen was noted when cells were exposed to nutrient-deprived conditions, in particular, BG_11_-N and BG_11_-NP (Figure 7C).

The *pha* transcript levels, related to PHB biosynthesis, were upregulated under nutrient deprivation, except for the *phaE* transcript under the BG_11_-NP condition, compared to those under the normal BG_11_ condition (Figure 8). For membrane lipid synthesis, *plsX* and *plsC* transcript levels were increased by BG_11_-N and BG_11_-P treatments, whereas they were downregulated under the BG_11_-NP condition. On the other hand, the increased *glgX* transcript levels, which involve glycogen degradation, were noted in all nutrient-deprived treatments when compared to those under the normal BG_11_ condition.

## 3. Discussion

In this study, we highlight the pragmatic finding of increased PHB production in the cyanobacterium *Synechocystis* sp. PCC 6803 overexpressing native *RuBisCO* genes and the relationship with lipid and glycogen metabolism (Figure 1). The primary CBB cycle converts atmospheric CO_2_ into carbon skeletons that are crucial for the production of valuable biocompounds. Recently, either native or heterologous expression of *RuBisCO* in cyanobacteria and higher plants was shown to cause increased growth and biomass yield, as well as increased photosynthesis efficiency [31,32,33]. Here, we introduced native *RuBisCO* or *rbc* genes, including *rbcL* and *rbcLXS* into *Synechocystis*, resulting in *rbcL*- and *rbcLXS*-overexpressing strains (OX*rbcL* and OX*rbcLXS*, respectively). Although we did not monitor the RuBisCO protein or its activity, the two OX strains had oxygen evolution rates that were 118–125 % higher than those of either WT or WTc under normal growth conditions (Figure 2E). However, it is worth noting that the post-translational regulation at the RuBisCO level may not be expressed in accordance with its transcriptional regulation in an overexpressing strain [33]. On the other hand, we discovered that the addition of NaHCO_3_ (20 mM) significantly accelerated the growth rate of all strains, resulting in the mid-long phase emerging after 3 days of culture compared to 5 days in the typical BG_11_ medium (Figure 3A,B). A low amount of NaHCO_3_ in a range of 5–20 mM concentrations was recommended for enhancing cyanobacterial growth due to excess NaHCO_3_ causing culture alkalinity [30,34,35]. When we measured chlorophyll *a* and carotenoids contents, there were no significant differences between WTc and OX strains grown in the normal BG_11_ medium (Figure 3C,E). The observed differences in these two pigments among all strains occurred in sodium bicarbonate-supplemented conditions, particularly after day 9 of culture, when the growth phase had reached the stationary phase (Figure 3D,F). These results suggested that a lower cell growth of WTc during the stationary phase in sodium bicarbonate-supplemented conditions in comparison with OX strains was partly caused by lower intracellular pigment accumulations. For PHB production, it was discovered that, although cells could produce PHB throughout the growth phase, the amount of PHB accumulated was only increased when cells entered the late-log and early-stationary phases of growth, either at day 11 under the normal condition or day 7 under the sodium bicarbonate condition (Figure 4A,B). The PHB contents were from 6–8 %w/DCW, with OX strains having higher levels than WTc. Moreover, an increased accumulation of total lipids was noted in all strains caused by sodium bicarbonate supplementation at day 3 and day 7 of cultivation (Figure 4C,D). It is important to point out that, in comparison to the condition for normal growth, glycogen accumulation was reduced in all strains under the sodium bicarbonate condition (Figure 4E,F). This result showed that *RuBisCO* overexpression had little impact on glycogen levels compared to WTc under both normal and sodium bicarbonate conditions, while the flow of important intermediates from the CBB cycle under sodium bicarbonate conditions preferentially reached acetyl-CoA, a precursor for the TCA cycle, fatty acid synthesis, and PHB production. Accordingly, a strategy to increase metabolic flow towards acetyl-CoA level in order to promote fatty acid synthesis and PHB accumulation was previously carried out either by imposing environmental challenges, such as an N-starved condition [23,36], or by genetic manipulation, such as through heterologous expression of the *xfpk* gene, encoding phosphoketolase, which catalyzes the cleavage of xylulose 5-phosphate (Xu5P) or fructose 6-phosphate (F6P) to acetyl-P and glyceraldehyde 3-phosphate or erythrose 4-phosphate [37,38].

To obtain high levels of PHB in cyanobacteria, a combination strategy comprising genetic alteration and environmental challenges seems more promising than a single strategy [23,24,26]. In this study, the adaptation of cultivation to nutrient starvation was practically applied for inducing PHB production (Figure 5). Although the growth rates of OX strains were higher than that of WTc under normal growth condition, no striking increases were found under all the nutrient starvation conditions studied. In contrast, an increase in PHB level was substantially induced by all nutrient-starved conditions, in particular BG_11_-N and BG_11_-NP treatments (Figure 6 and Figure 9). Under these BG_11_-N and BG_11_-NP conditions, the OX*rbcLXS* strain exhibited the highest level of PHB by approximately 38.1–39.7 %w/DCW at days 7–9 of treatment (Figure 6). A high PHB content of between about 34.5 and 34.6 %w/DCW was similarly produced by OX*rbcL* on day 9 of the BG_11_-N and BG_11_-NP treatments. Chlorotic cells caused by nitrogen deprivation accumulate glycogen and PHB as carbon sources for cellular acclimation and survival [23,39,40,41], activated by the nitrogen regulators NtcA and PirC, DNA-transcriptional activators, which consequently trigger physiological changes, such as growth restriction, degradation of phycobilisomes, and chlorophyll *a* reduction [42,43,44,45,46]. Glycogen, which serves as the major carbon source, could subsequently contribute to increased PHB production during nitrogen depletion [23,46]. Our findings demonstrate that the amount of glycogen in WTc increased by 8- to 9-fold throughout 7 days of N- and NP-starvation compared to that under normal BG_11_, but the induced glycogen in OX*rbcL* and OX*rbcLXS* exhibited lower content than WTc (Figure 7C). On the other hand, the phosphorus deprivation gradually induced a slight increase in glycogen and PHB throughout the 9 days of treatment (Figure 6C and Figure 7C). In contrast to WTc, it was interesting to find out that high PHB production started in both OX strains on day 9 of P-starved treatment (Figure 6C). During the initial 7 to 9 days of the BG_11_-NP condition, the minor synergistic effect of P-starvation was shown to increase PHB content (Figure 6D). Phosphorus deprivation lowers ATP production, while noncyclic photosynthetic electrons continuously flow and yield an increased NADPH pool, which is beneficial in PHB synthesis [47,48].

In Figure 9, a summary of WTc and all engineered strains is shown under all nutrient-deprived conditions for 7 days. Glycogen and PHB accumulation in WTc were unquestionably raised by approximately 8- to 9-fold under N- and NP-starvation compared to that under the normal BG_11_ condition. It is important to note that *RuBisCO* gene overexpression at day 7 of treatment under both N- and NP-starvation conditions drastically lowered glycogen content but increased PHB content. This phenomenon was confirmed by the 12- to 15-fold increase in *glgX* transcript levels in the OX*rbcL* and OX*rbcLXS* strains compared to those under the normal condition, which are implicated in higher glycogen breakdown. In addition, *phaA*, *phaB*, and *phaC* transcript levels increased more than 8-fold in response to N- and NP-deprivation, but not the *phaE* transcript, which was dominant in increasing PHB synthesis in OX strains. In contrast, the total and unsaturated lipid levels did not change substantially during N- and NP-starvation, and cells responded to N-stress by preferentially maintaining intracellular lipid homeostasis. In contrast, phosphorus (P) deprivation raised PHB by 4-fold compared to that under the normal BG_11_ condition but had no effect on glycogen deposition in *RuBisCO-*overexpressing strains compared to WTc. The *phaA* and *phaC* transcript levels were highly upregulated by more than 4- to 10-fold by P-stress when compared to the normal BG_11_ condition, whereas the *phaB* and *phaE* transcript levels were only slightly altered.

## 4. Materials and Methods

### 4.1. Construction of rbc-Overexpressing Synechocystis sp. PCC 6803

Initially, the recombinant plasmids, including pEERM_*rbcL* and pEERM_*rbcLXS* (Table 1), were constructed to generate all overexpressing (OX) strains, including *rbcL*- and *rbcLXS*-overexpressing *Synechocystis* strains (OX*rbcL* and OX*rbcLXS*, respectively). The construction of the pEERM_*rbcL* and pEERM_*rbcLXS* was performed by ligating the *rbcL* and *rbcLXS* genes amplified by PCR using a pair of rbcL_F and rbcL_R primers and another pair of rbcL_F and rbcS_R primers, respectively, as listed in Appendix A, in between the restriction sites of *SpeI* and *PstI* in the pEERM vector [29]. After that, the recombinant plasmids were confirmed by double digestion with the *SpeI* and *PstI* restriction enzymes. Therefore, the correct recombinant plasmids (pEERM_*rbcL* and pEERM_*rbcLXS*) were transformed into *Synechocystis* sp. PCC 6803 WT cells by natural transformation, thereby generating OX*rbcL* and OX*rbcLXS*, respectively. For construction of the *Synechocystis* sp. PCC wild type control (WTc), the empty pEERM vector was transformed into *Synechocystis* WT cells, represented as *Synechocystis* WT containing the *Cm^R^* cassette gene. For host cell suspension preparation, the *Synechocystis* sp. PCC 6803 host cells were cultured in BG_11_ medium until OD at 730 nm increased by about 0.3–0.5. Then, the cell culture (50 mL) was harvested by centrifugation at 5500 rpm (3505× *g*), 25 °C for 10 min, and cell pellets were resuspended in fresh BG_11_ medium (500 µL). After this, at least 10 µL of constructed recombinant plasmids were added and mixed with the *Synechocystis* host cell suspension and incubated in the culture room with continuous white light illumination at 27–30 °C for 16–18 h. Then, the sample mixture was spread on a BG_11_ agar plate containing 10 µg/mL of chloramphenicol. All agar plates were incubated in the culture room until surviving colonies occurred (for about 2–3 weeks). Each single colony was picked and streaked on a new BG_11_ agar plate containing chloramphenicol up to 20 and 30 µg/mL concentration. The obtained transformants were confirmed for gene size, location, and segregation by a PCR method using many specific pairs of primers, as shown in Appendix A.

### 4.2. Strains and Culture Conditions

*Synechocystis* sp. PCC 6803 wild type (WT) was derived from the Berkeley strain 6803, which was isolated from fresh water in California, USA [2], and all engineered strains (WTc, OX*rbcL*, and OX*rbcLXS*) were grown in normal BG_11_ medium and BG_11_ medium supplemented with 20 mM NaHCO_3_. The normal growth condition was performed at 27–30 °C, under white light illumination with 40–50 μmole photon/m^2^/s intensity. The culture flask with an initial cell density at 730 nm (OD730) of about 0.1 was placed on a shaker (160 rpm). Cell growth was determined by spectrophotometrically measuring OD730. For the nutrient-deprived treatment, all *Synechocystis* strains were initially grown in BG_11_ containing 20 mM NaHCO_3_ for 7 days before treating the harvested cell pellets with nutrient starvation. The nutrient-deprived media included (1) BG_11_ without NaNO_3_ with added FeSO_4_ instead of ferric ammonium citrate (BG_11_-N), (2) BG_11_ with KCl addition instead of KH_2_PO_4_ (BG_11_-P), and (3) BG_11_ without nitrogen and phosphorus (BG_11_-NP). The initial OD730 of the cell culture under nutrient-stressed treatment was adjusted to about 0.45.

### 4.3. Determinations of Intracellular Pigments and Oxygen Evolution Rate

One mL of cell culture was harvested by centrifugation at 5500 rpm (3505× *g*) for 10 min. The intracellular chlorophyll *a* and carotenoids in the cell pellets were extracted by N,N-dimethylformamide (DMF), vortexed, and incubated in the dark for 10 min. After quickly spinning to remove debris, the absorbance of the yellowish supernatant was spectrophotometrically measured for its absorbances at 461, 625 and 664 nm. The contents of chlorophyll *a* and carotenoids were calculated according to [49,50]. The photosynthetic efficiency was determined in terms of the oxygen evolution rate. The cell culture (5–10 mL) was harvested by centrifugation at 5500 rpm (3505× *g*) for 10 min. Cell pellets were resuspended in fresh BG_11_ medium (1 mL) and incubated under darkness for 30 min before measuring the oxygen evolution rate. Saturated white light was used as a light source at 25 °C using a Clark-type oxygen electrode (Hansatech instruments Ltd., King’s Lynn, UK). The unit of oxygen evolution rate was represented as µmol O_2_/mg chlorophyll *a*/h [24].

### 4.4. Total RNAs Extraction and Reverse Transcription-Polymerase Chain Reaction (RT-PCR)

Total RNAs were isolated from each strain using the TRIzol^®^ Reagent (Invitrogen, Life Technologies Corporation, Carlsbad, CA, USA). The purified RNAs (1 µg) were converted to cDNA by reverse transcription using ReverTra ACE^®^ qPCR RT Master Mix Kit (TOYOBO Co., Ltd., Osaka, Japan). Then, the cDNA was further used as the template for PCR amplification with different pairs of primers, as listed in Appendix A. The PCR conditions were performed by initial denaturing at 94 °C for 3 min, followed by suitable cycles of each gene (Appendix A) at 94 °C for 30 s, primer melting temperature (Tm, Appendix A) for 30 s, and 72 °C for 20 s, and then final extension at 72 °C for 7 min. After that, the PCR products were checked by 1% (*w*/*v*) agarose gel electrophoresis. Quantification of band intensity was determined by Syngene^®^ Gel Documentation (Syngene, Frederick, MD, USA).

### 4.5. Quantitative Analysis of PHB Contents

The PHB content was detected by high-performance liquid chromatography (HPLC) (Shimadzu HPLC LGE System, Kyoto, Japan). Cell cultures (15–30 mL) were harvested by centrifugation at 5500 rpm (3505× *g*) for 10 min. To prepare samples, 100 µL of internal standard (20 mg/mL adipic acid) and 800 µL of 98% (*v*/*v*) sulfuric acid were added into cell pellets. After mixing, the sample mixture was boiled at 100 °C for 1 h for digesting PHB to crotonic acid (modified from [24]). Then, the reaction mixture was filtered using a 0.45 µm polypropylene membrane filter, and detected using a carbon-18 column, Inert Sustain 3-µm (GL Sciences, Tokyo, Japan) and UV detector at 210 nm. The running buffer was 30% (*v/v*) acetonitrile in 70% (*v/v*) of 10 mM KH_2_PO_4_ (pH 2.3), with a flow rate of about 1.0 mL/minute. Authentic commercial PHB (Sigma-Aldrich, Inc., St. Louis, MO, USA) was used as standard, which was prepared using the same method as the samples. In addition, the dry cell weight (DCW) was determined by incubating cell pellets at 60 °C until obtaining a constant weight [26].

### 4.6. Lipid Extraction and Determinations of Total Lipid and Unsaturated Lipid Contents

Cell culture (25 mL) was harvested by centrifugation at 5500 rpm (3505× *g*) for 10 min. The obtained cell pellet was mixed with 1 mL of CHCl_3_:MeOH (2:1 ratio) solution and incubated at 55 °C for 2 h. Then, distilled water (0.5 mL) was added and mixed by vortexing for 2 min. After leaving at room temperature for 10 min, the sample mixture was centrifuged at 5500 rpm (3505× *g*) for 10 min. The chloroform phase (lower layer) containing lipids was collected, while the upper aqueous phase was re-extracted by addition of chloroform (500 µL) (modified from [12,51]). The lipid-containing chloroform phase was pooled and evaporated in a fume hood at room temperature overnight to obtain total lipids. The total lipid content was determined by an acid-dichromate oxidation method (modified from [28,52]. The lipid extract was dissolved in 1 mL of 98% H_2_SO_4_ and later mixed by vortexing. After that, 1 mL of potassium dichromate (K_2_CrO_7_) solution was added to the sample and boiled at 100 °C for 30 min, followed by cooling down to room temperature. Then, 1 mL of distilled water was added and mixed. The sample was measured for absorbance at 600 nm by a spectrophotometer. Commercial canola oil was used as standard. The unit of total lipid content was %w/DCW.

The unsaturated lipid content was measured using the colorimetric sulfo-phospho-vanilin (SPV) method, with commercial linoleic acid serving as the standard (modified from [53]). Extracted lipids were dissolved in 1 mL of 98% H_2_SO_4_ and shaken vigorously before boiling at 100 °C for 30 min. After that, the boiled samples were cooled down to room temperature. One mL of a solution of 17% (*v*/*v*) H_3_PO_4_ and 0.2 mg/mL vanillin was added and incubated for 10 min at room temperature. The absorbance at 540 nm was spectrophotometrically measured to determine the unsaturated lipid content. The unit of total unsaturated lipid content was %w/DCW.

### 4.7. Glycogen Extraction and Determination of Glycogen Content

Glycogen was extracted by alkaline hydrolysis [54,55]. The cell culture (15–30 mL) was harvested by centrifugation at 5500 rpm (3505× *g*), at 25 °C for 10 min. The cell pellets were resuspended in 400 µL of 30 %(*v/v*) KOH and boiled at 100 °C for 1 h. After centrifugation at the same speed to collect the supernatant, 900 µL of cold absolute ethanol was added and then incubated at −20 °C overnight to precipitate glycogen. To obtain the glycogen pellets, the sample mixture was centrifuged at 12,000 rpm (14,489× *g*), at 4 °C for 30 min, and dried at 60 °C for overnight. The anthrone assay was used for determining the glycogen content, and commercial oyster glycogen (Sigma-Aldrich, St. Louis, MO, USA) served as the standard (modified from [56,57]). Glycogen pellets were dissolved with 1 mL of 10% H_2_SO_4_. Then, 0.2 mL of dissolved sample was mixed with 0.2 mL of 10% H_2_SO_4_ and 0.8 mL of anthrone reagent. The reaction mixture was then boiled at 100 °C for 10 min. After cooling down the samples to room temperature, their absorbance at 625 nm was measured by spectrophotometer. The unit of glycogen content was %w/DCW

## 5. Conclusions

Enhanced levels of PHB accumulation were attained in engineered strains of *Synechocystis* sp. PCC 6803 (OX*rbcL* and OX*rbcLXS*, involved in CBB cycle) under nutrient-deprived conditions. A faster growth rate was found in OX strains under the normal growth condition. Following the application of N- and NP-stresses, a significant carbon flux from glycogen storage flowing to PHB synthesis was definitely triggered as a stress acclimation response. In Figure 9, the mechanistic perspective provided indicates that *RuBisCO* overexpression responded to nutrient deprivation, in particular N- and NP-stresses, by driving cells to accumulate PHB rather than glycogen and lipids when compared to wild type control cells, which preferred to store more glycogen than PHB and lipids. These *RuBisCO*-overexpressing *Synechocystis* strains are potential cell factories for the production of value-added chemicals, such as biomaterials and biofuels.

## Figures and Tables

**Figure 1 ijms-24-06415-f001:**
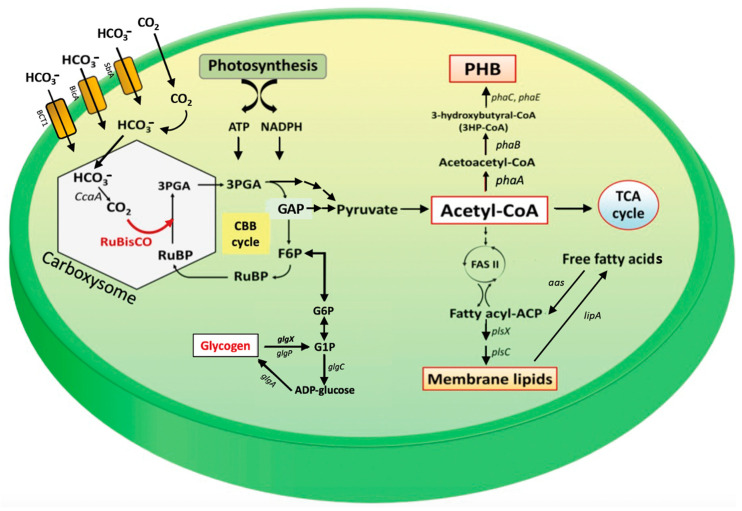
Overview of the Calvin–Benson–Bassham (CBB) cycle and related biosynthetic pathways of polyhydroxybutyrate (PHB), membrane lipids, and glycogen in cyanobacterium *Synechocystis* sp. PCC 6803. Abbreviations of genes are: *aas*, acyl-ACP synthetase; *ccaA*, carbonic anhydrase; *glgA*, glycogen synthase; *glgC*, ADP-glucose pyrophosphorylase; *glgP*, glycogen phosphorylase; *glgX*, glycerol-3-phosphate dehydrogenase; *lipA*, a lipolytic enzyme-encoding gene; *phaA*, β-ketothiolase; *phaB*, acetoacetyl-CoA reductase; *phaE* and *phaC*, the heterodimeric PHB synthase; *plsX* and *plsC*, putative phosphate acyl-transferases; *RuBisCO*, the RuBisCO gene cluster, including *rbcLXS*, encoding RuBisCO large, chaperone, and small subunits, respectively. Abbreviations of intermediates are: FASII, fatty acid synthesis type II; fatty acyl-ACP, fatty acyl–acyl carrier protein; G1P, glucose 1-phosphate; F6P, fructose 6-phosphate; G6P, glucose 6-phosphate; GAP, glyceraldehyde-3-phosphate; 3PGA, 3-phosphoglycerate; PHB, polyhydroxybutyrate; RuBP, ribulose-1,5-bisphosphate; TCA cycle, tricarboxylic acid cycle.

**Figure 2 ijms-24-06415-f002:**
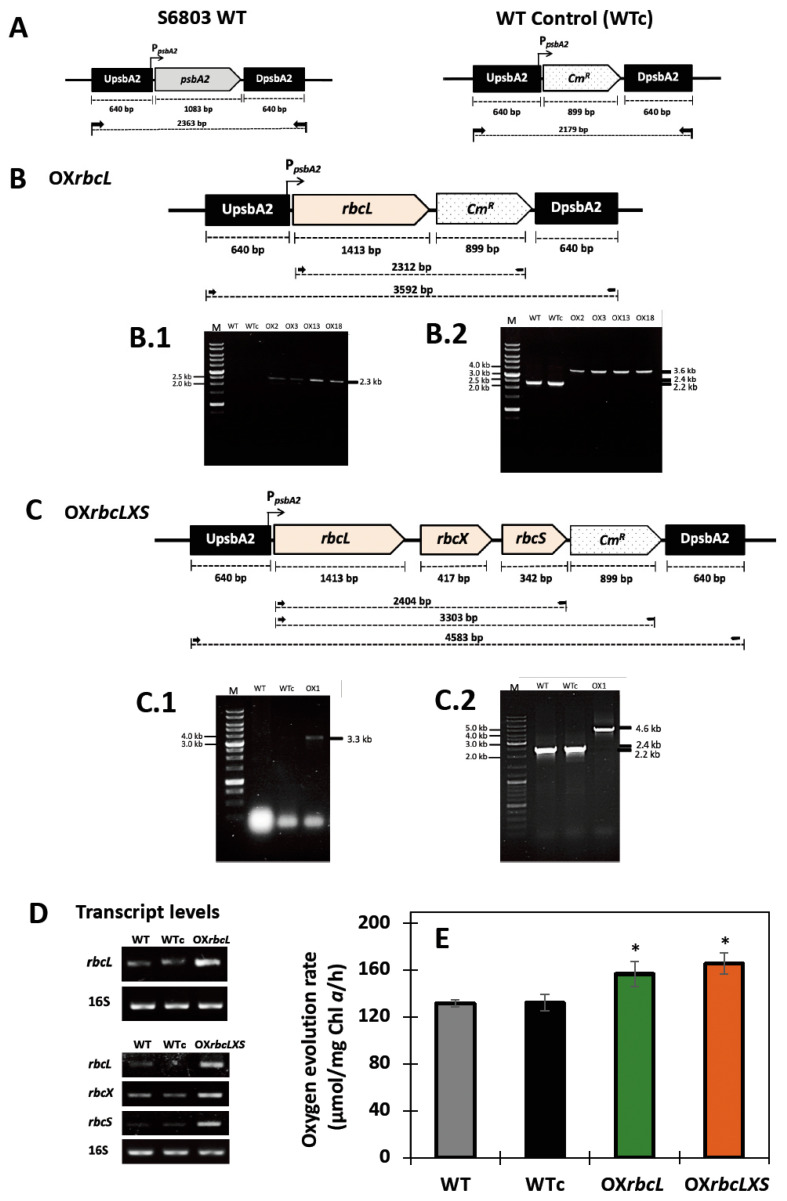
Genomic maps, transcript levels, and oxygen evolution rates of *Synechocystis* sp. PCC 6803 strains. The three engineered strains are (**A**) wild type control (WTc), (**B**) OX*rbcL*, and (**C**) OX*rbcLXS*. Confirmations of the corrected integration and location of each gene fragment into the *Synechocystis* genome were performed by PCR analysis using various pairs of specific primers (shown in Appendix A). (**A**) The double homologous recombination of *Cm^R^* gene occurred between the conserved sequences of *psbA2* gene in WTc when compared to WT. For (**B**) OX*rbcL* strain; Lane M: GeneRuler DNA ladder (Fermentus); (**B.1**) PCR products using rbcL_F and CM_R primers, Lanes WT and WTc: Negative control using WT and WTc as template, respectively, Lanes OX2, OX3, OX13, and OX18: four clones no. 2, 3, 13, and 18 containing a 2.3 kb fragment of *rbcL*-*Cm^R^*; (**B.2**) PCR products using UppsbA2_F and DSpsbA2_R primers, Lanes WT and WTc: Negative controls of a 2.4 kb fragment in WT and a 2.2 kb fragment in WTc, respectively, Lanes OX2, OX3, OX13, and OX18: four clones no. 2, 3, 13, and 18 containing a 3.6 kb fragment of *UpsbA2*-*rbcL*-*Cm^R^*-*DpabA2*. For (**C**) OX*rbcLXS* strain; Lane M: GeneRuler DNA ladder (Fermentus); (**C.1**) PCR products using rbcL_F and CM_R primers, Lanes WT and WTc: Negative control using WT and WTc as template, respectively, Lanes OX1: one clone no. 1 containing a 3.3 kb fragment of *rbcL-rbcX-rbcS-Cm^R^*; (**C.2**) PCR products using UppsbA2_F and DSpsbA2_R primers, Lanes WT and WTc: Negative controls of a 2.4 kb fragment in WT and a 2.2 kb fragment in WTc, respectively, Lane OX1: one clone no. 1 containing a 4.6 kb fragment of *UpsbA2-rbcL-rbcX-rbcS-Cm^R^-DpabA2*. Transcript levels (**D**) measured by RT-PCR using specific primers, shown in Appendix A, of *rbc* genes in WT, WTc and two overexpressing strains, including OX*rbcL* and OX*rbcLXS*. The 0.8% agarose gel electrophoresis of PCR products of all transcripts was performed from cells grown for 7 days in normal BG_11_ medium. The *16s* rRNA was used as reference. Oxygen evolution rates (**E**) of *Synechocystis* sp. PCC 6803 WT, WTc, OX*rbcL* and OX*rbcLXS* cells grown in normal BG_11_ medium condition for 3 days (means ± S.D., n = 3). The statistical difference in the data between the values of WT and the engineered strain is represented by an asterisk at * *p* < 0.05. The cropped gels (in (**D**)) were taken from the original images of RT-PCR products on agarose gels as shown in Appendix A).

**Figure 3 ijms-24-06415-f003:**
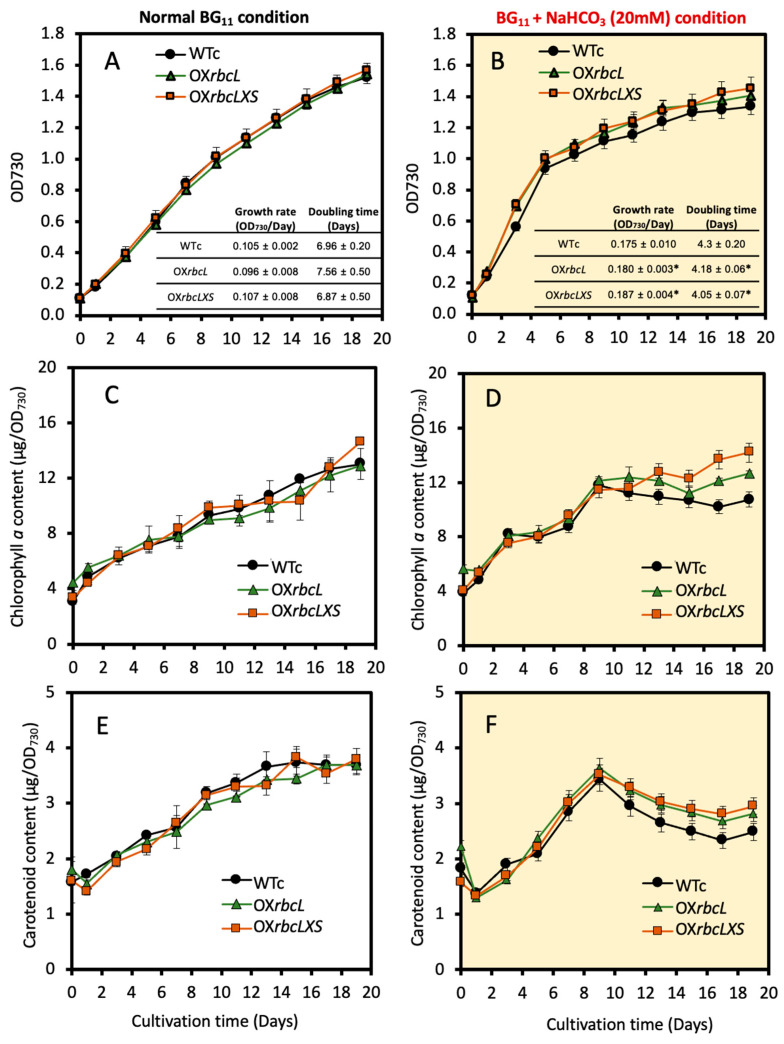
Optical density at 730 nm (**A**,**B**), chlorophyll *a* contents (**C**,**D**), and carotenoid contents (**E**,**F**) of *Synechocystis* WTc, OX*rbcL*, and OX*rbcLXS* strains grown in normal BG_11_ medium and BG_11_ medium containing 20 mM NaHCO_3_ (BG_11_ + NaHCO*_3_* condition), respectively, for 19 days. In (**A**) and (**B**), the growth rates and doubling times are provided as table insets. The statistical difference of the data between the values of WT and the engineered strain is represented by an asterisk at * *p* < 0.05. The error bars represent standard deviations of means (means ± S.D., n = 3).

**Figure 4 ijms-24-06415-f004:**
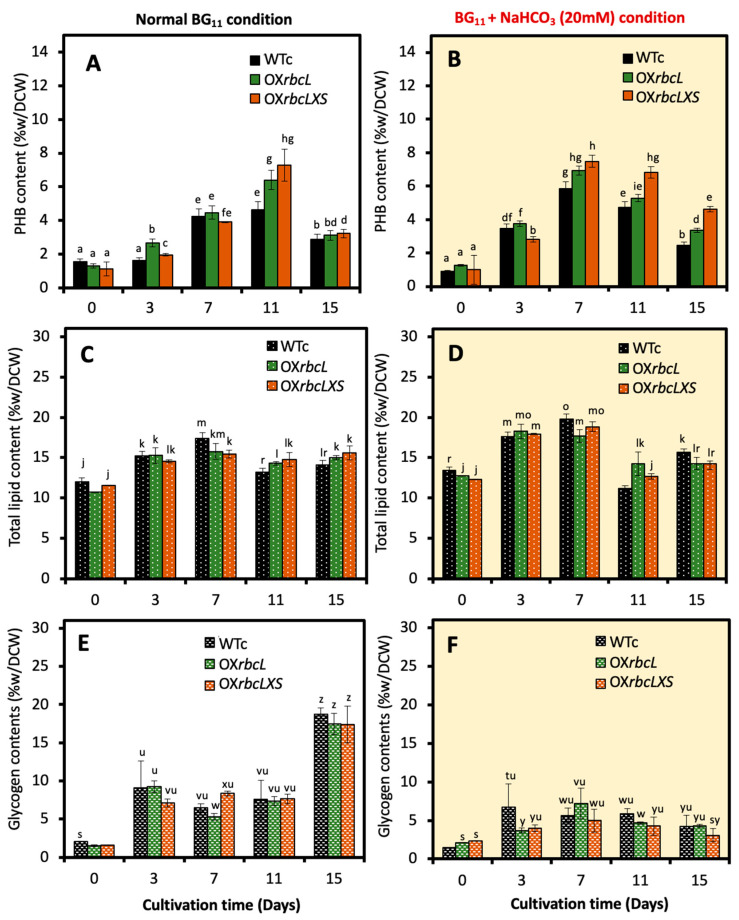
PHB contents (**A**,**B**), total lipid contents (**C**,**D**), and glycogen contents (**E**,**F**) of *Synechocystis* WTc, OX*rbcL*, and OX*rbcLXS* strains grown in normal BG_11_ medium and BG_11_ medium containing 20 mM NaHCO_3_ (BG_11_ + NaHCO*_3_* condition), respectively, for 15 days. The error bars represent standard deviations of means (means ± S.D., n = 3). Means with the same letter are not significantly different with the significance level at *p* < 0.05.

**Figure 5 ijms-24-06415-f005:**
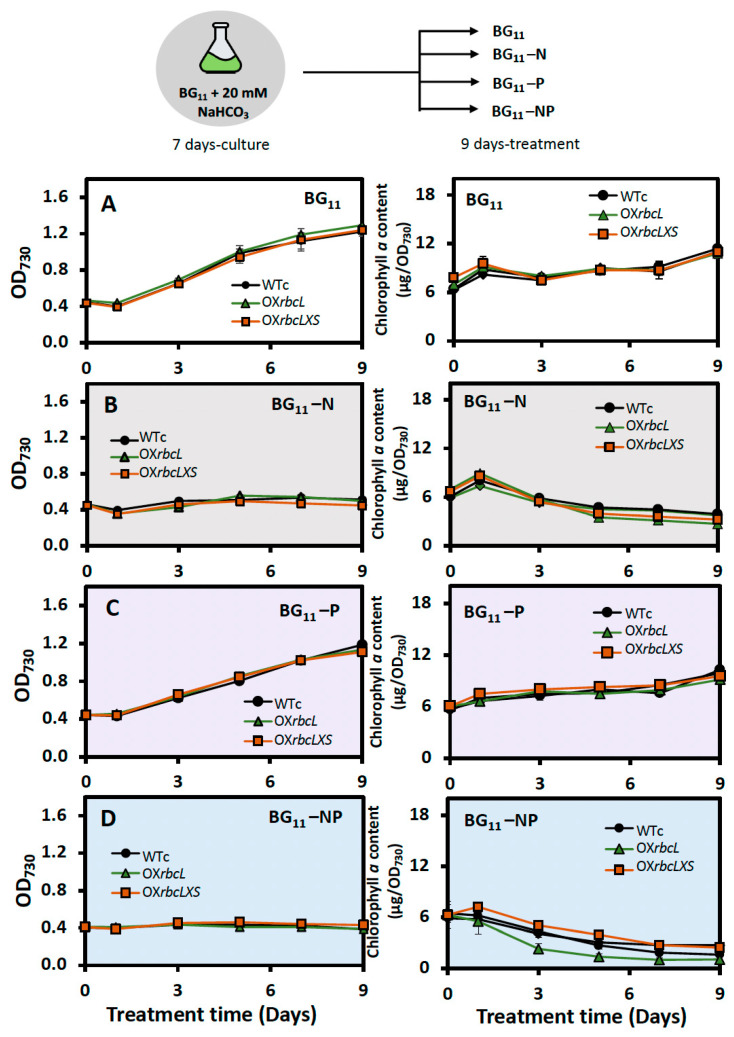
Optical density at 730 nm and chlorophyll *a* content of *Synechocystis* WTc, OX*rbcL*, and OX*rbcLXS* strains after adapting in (**A**) normal BG_11_ medium, (**B**) BG_11_-N, (**C**) BG_11_-P, and (**D**) BG_11_-NP for 9 days. The error bars represent standard deviations of means (means ± S.D., n = 3).

**Figure 6 ijms-24-06415-f006:**
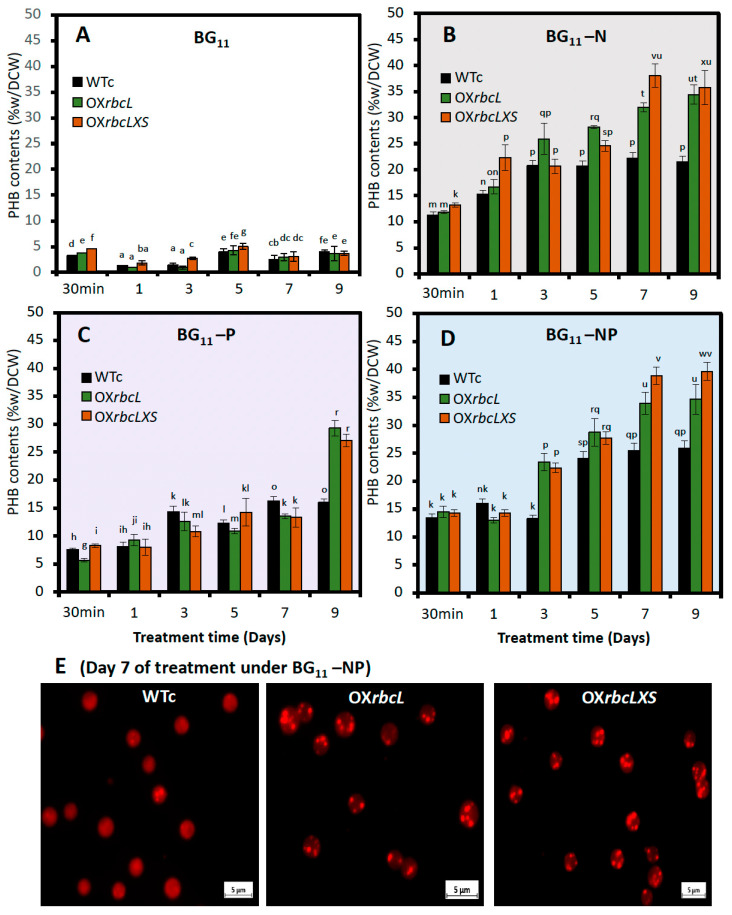
PHB contents of *Synechocystis* WTc, OX*rbcL*, and OX*rbcLXS* strains after adapting in (**A**) normal BG_11_ medium, (**B**) BG_11_-N, (**C**) BG_11_-P, and (**D**) BG_11_-NP for 9 days. The error bars represent standard deviations of means (means ± S.D., n = 3). Means with the same letter are not significantly different with the significance level at *p* < 0.05. (**E**) The Nile red staining of neutral lipids, herein PHB granules, in all strains treated by BG_11_-NP condition for 7 days. The stained cells were visualized under fluorescent microscope with a magnification of ×100.

**Figure 7 ijms-24-06415-f007:**
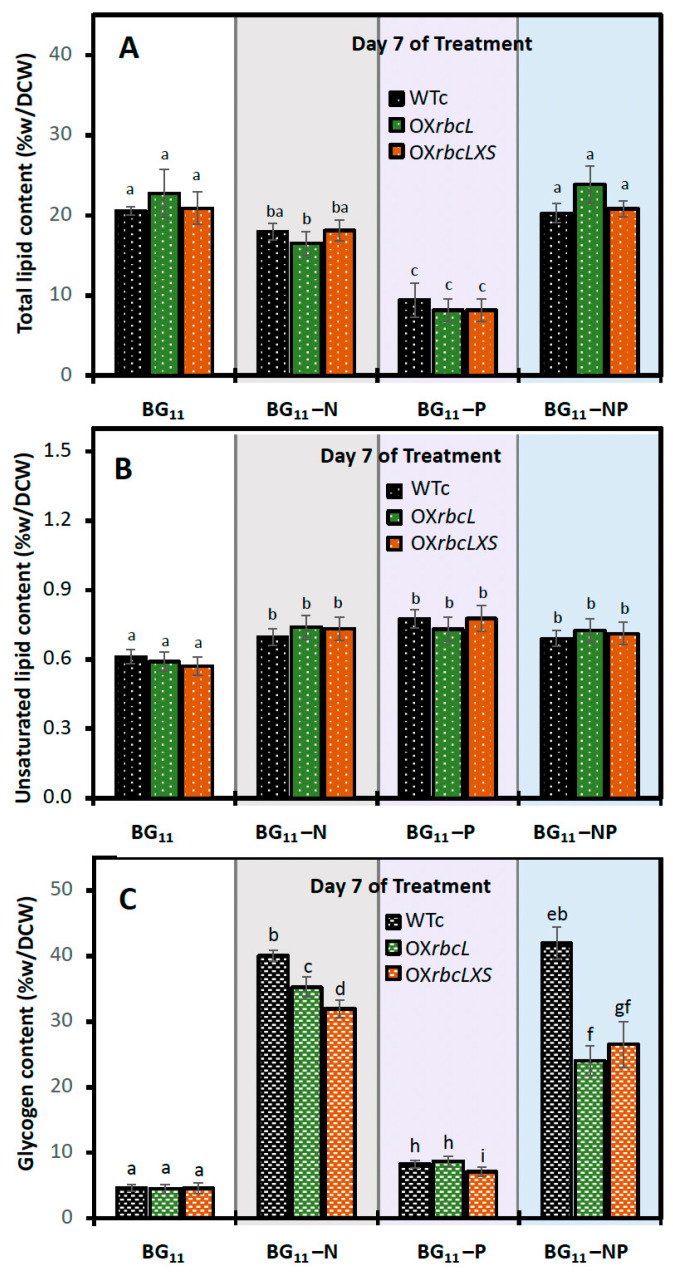
Contents of (**A**) total lipids, (**B**) unsaturated lipids, and (**C**) glycogen in *Synechocystis* WTc, OX*rbcL*, and OX*rbcLXS* strains after adapting in normal BG_11_ medium, BG_11_-N, BG_11_-P, and BG_11_-NP for 7 days. The error bars represent standard deviations of means (means ± S.D., n = 3). Means with the same letter are not significantly different with the significance level at *p* < 0.05.

**Figure 8 ijms-24-06415-f008:**
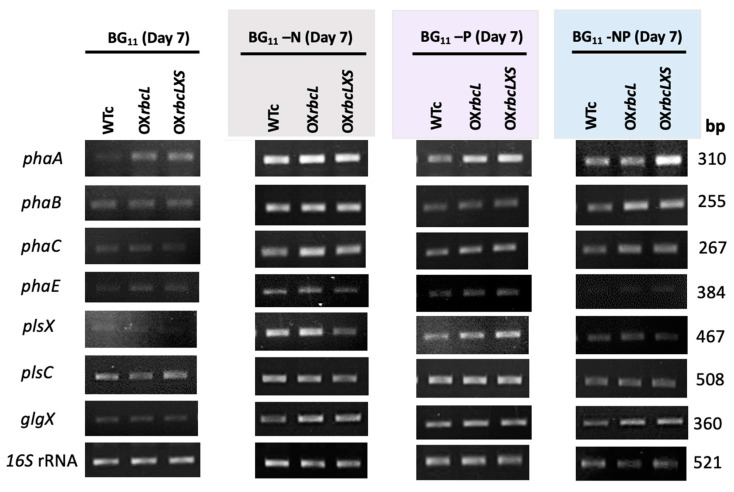
Relative transcript levels of *phaA*, *phaB*, *phaC*, *phaE*, *plsX*, *plsC*, and *glgX* performed by RT-PCR in *Synechocystis* WTc, OX*rbcL*, and OX*rbcLXS* strains after adapting in normal BG_11_ medium, BG_11_-N, BG_11_-P, and BG_11_-NP for 7 days. The *16s* rRNA was used as reference control. The band intensity data were shown in Appendix A. All cropped gels were taken from the original images of RT-PCR products on agarose gels as shown in Appendix A).

**Figure 9 ijms-24-06415-f009:**
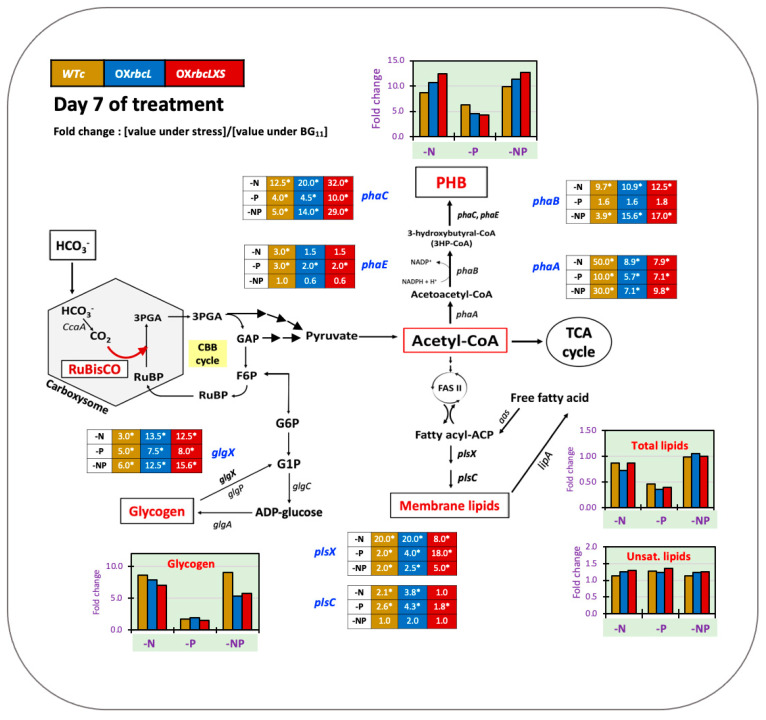
Summary of obtained results, including products and gene transcript levels from two overexpressing strains, including comparisons to *Synechocystis* sp. PCC6803 WTc after adapting cells in BG_11_, BG_11_-N, BG_11_-P, and BG_11_-NP at day 7 of treatment. In each box and bar graph, the number and bar represent the fold change of that value of each strain under stress condition divided by that value of that strain under normal BG_11_ condition. The statistical difference in the data between those values of WT and the engineered strain is represented by an asterisk at * *p* < 0.05.

**Table 1 ijms-24-06415-t001:** Strains and plasmids used in this study.

Name	Relevant Genotype	Reference
Cyanobacterial strains		
*Synechocystis* sp. PCC 6803	Wild type	Pasteur culture collection
WT control	WT, *Cm^R^* integrated at flanking region of *psbA2* gene in *Synechocystis* genome	This study
OX*rbcL*	*rbcL*, *Cm^R^* integrated at flanking region of *psbA2* gene in *Synechocystis* genome	This study
OX*rbcLXS*	*rbcLXS*, *Cm^R^* integrated at flanking region of *psbA2* gene in *Synechocystis* genome	This study
**Plasmids**		
pEERM	P_psbA2_- *Cm^R^*; plasmid containing flanking region of *psbA2* gene	[29]
pEERM_*rbcL*	P_psbA2_-*rbcL*- *Cm^R^*; integrated between *Spe*I and *Pst*I sites of pEERM	This study
pEERM_*rbcLXS*	P_psbA2_-*rbcLXS*- *Cm^R^*; integrated between *Spe*I and *Pst*I sites of pEERM	This study

P_psbA2_, *psbA2* promoter; *Cm^R^*, chloramphenicol resistance cassette.

## Data Availability

Not applicable.

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
