# Peer review of "Increased Biomass and Polyhydroxybutyrate Production by Synechocystis sp. PCC 6803 Overexpressing RuBisCO Genes"

_ijms, 2023, doi:10.3390/ijms24076415_

Round 1

Reviewer 1 Report

In this study, the authors successfully constructed two rbc-overexpressing Synechocystis sp. PCC 6803 strains (OXrbcL and OXrbcLXS), resulting higher and faster growth than wild type under bicarbonate supplementation, and significantly contributed to the higher PHB production induced by nutrient-deprived conditions. The experiment design is reasonable, the logic is clear, and the writing is good. A minor problem, abbreviations must be defined at their first mention, for example, line 15 “PHB”, line 19 “DCW”. Another question, how do we balance cyanobacterial biomass and polyhydroxybutyrate production, since the authors said that cyanobacteria growth and biomass can be promoted under normal or bicarbonate supplementation conditions, while PHB synthesis can be promoted under nutrient restriction conditions.

Reviewer 2 Report

Dear Corresponding Author

I checked your paper and I think you need to add some more relevant references into Introduction and correct minor English errors throughout the text.

Regards

Reviewer 3 Report

The manuscript ID ijms-2292806 describes two rbc-overexpressing strains involving high and fast growth under bicarbonate supplementation and a high polyhydroxybutyrate production induced by nutrient-deprived conditions. The manuscript is interesting and includes relevant information for readers. However, some points should be addressed before being considered further.

1.      Detailed scrutiny should be performed throughout the manuscript to look for some grammar, stylistic, and even typos issues.

2.      Line 15: The abbreviation PHB should be.

3.      Line 19: Define DCW.

4.      Abstract: Define all abbreviations used for the first time in the abstract. Be consistent in the manuscript.

5.      Figure 2. Improve the quality and size of this figure since it is challenging to visualize the information.

6.      Line 172: (sodium bicarbonate) instead of (bicarbonate) since the compound formula is related to the entire salt. If the carbon source is bicarbonate, and sodium is a spectator counterion, then the formula should be HCO3-.

7.      Figure 3: Imrpove the size of this figure. In this regard, be consistent with the size of all figures. For instance, Figures 3 and 4 have relevant size and quality differences.

8.      Line 204: revise the numbers associated with multiple comparisons since they appeared to have misleadings. For instance, the letter sequences and groupings seemed to be not correct.

9.      Line 373: The origin of the wild Synechocystis sp. strain must be informed.

10.   Line 441: More details about PHB quantification are missing. For instance, the analytical quality parameters should be added, such as recovery, LOD, LOQ, etc, to ensure reproducibility. In addition, the method for quantification is missing (e.g., internal or external standard), and such conditions and parameters must be provided.

11.   The conclusion section can be improved since there is a brief summary of results, but conceptual findings from the mechanistic point of view should be provided (e.g., from figure 9).
